# Effectiveness of integrated chronic care models for cardiometabolic multimorbidity in sub-Saharan Africa: a systematic review and meta-analysis

Peter Otieno [1,2,3] Charles Agyemang,[2] Hesborn Wao [4] Elvis Wambiya [5,6] Maurine Ng'oda,[7] Daniel Mwanga,[1] James Oguta,[5,6] Peter Kibe,[1,6] Gershim Asiki[1,8]

For numbered affiliations see end of article.

**Correspondence to**
Peter Otieno;
pootienoh@gmail.com

## ABSTRACT

**Objectives** This review aimed at identifying the elements of integrated care models for cardiometabolic multimorbidity in sub-Saharan Africa (SSA) and their effects on clinical or mental health outcomes including systolic blood pressure (SBP), blood sugar, depression scores and other patient-reported outcomes such as quality of life and medication adherence.

**Design** Systematic review and meta-analysis using the Grading of Recommendation, Assessment, Development and Evaluation (GRADE) approach.

**Data sources** We systematically searched PubMed, Embase, Scopus, Web of Science, Global Health CINAHL, African Journals Online, Informit, PsycINFO, ClinicalTrials. gov, Pan African Clinical Trials Registry and grey literature from OpenSIGLE for studies published between 1999 and 2022.

**Eligibility criteria for selecting studies** We included randomised controlled trial studies featuring integrated care models with two or more elements of Wagner's chronic care model.

**Data extraction and synthesis** Two independent reviewers used standardised methods to search and screen included studies. Publication bias was assessed using the Doi plot and Luis Furuya Kanamori Index. Meta-analysis was conducted using random effects models.

**Results** In all, we included 10 randomised controlled trials from 11 publications with 4864 participants from six SSA countries (South Africa, Kenya, Nigeria, Eswatini, Ghana and Uganda). The overall quality of evidence based on GRADE criteria was moderate. A random-effects meta-analysis of six studies involving 1754 participants shows that integrated compared with standard care conferred a moderately lower mean SBP (mean difference=−4.85 mm Hg, 95% CI −7.37 to −2.34) for people with cardiometabolic multimorbidity; Hedges' g effect size (g=−0.25, (−0.39 to −0.11). However, integrated care compared with usual care showed mixed results for glycated haemoglobin, depression, medication adherence and quality of life.

**Conclusion** Integrated care improved SBP among patients living with cardiometabolic multimorbidity in SSA. More studies on integrated care are required to improve the evidence pool on chronic care models for multimorbidity in SSA. These include implementation studies and cost-effectiveness studies.

## STRENGTHS AND LIMITATIONS OF THIS STUDY

⇒ The inclusion of randomised controlled trials provides more unbiased findings than previous reviews that are mostly based on observational study designs.
⇒ The extraction of important constructs of Wagner's chronic care model from the integrated care models offers deeper insights into the effectiveness of integrated care model elements.
⇒ The review considered both communicable and non-communicable diseases which reflects their converging burdens in sub-Saharan Africa.
⇒ The varying numbers and types of chronic care model components resulted in heterogeneity in the included studies.
⇒ Since most included studies used multicomponent interventions, the absolute effect attributable to a particular element of the chronic care model remains unknown.

**PROSPERO registration number** CRD42020187756.

## BACKGROUND

Cardiometabolic diseases, including type 2 diabetes and cardiovascular diseases (CVDs), are the leading cause of global mortality.[1] People living with cardiometabolic diseases often have multiple rather than a single condition, commonly known as multimorbidity.[2] Cardiometabolic multimorbidity occurs against the background of obesity and insulin resistance.[3] However, these conditions are also characterised by the simultaneous co-occurrence of other diseases with unrelated pathophysiology or care management profile such as HIV/AIDS, chronic respiratory diseases, cancers and mental illnesses, a phenomenon referred to as discordant multimorbidity.[4–6]

The burden of multimorbidity is highest in low-income and middle-income countries (LMICs), which still bear a high burden of

infectious diseases while also facing a new threat from non-communicable diseases.[7] However, the majority of the chronic care models in sub-Saharan Africa (SSA) are still built around single disease frameworks.[8] Evidence shows that health services delivery based on isolated interventions may derail the value of treatment and preventive measures.[9–12] Vertical disease programmes result in missed opportunities for early diagnosis and management of multimorbidities.[13] Therefore, integrated chronic disease management is recommended to respond to this challenge.[14] Singer et al,[15] define *integrated care* as a set of patient-centred and multidisciplinary care activities coordinated by two or more collaborating service providers within or across the healthcare sector including community and social environments. This definition emphasises the coordination of care activities and the active involvement of patients in managing their health.

The chronic care model, first identified by McColl Institute for Healthcare Innovation at Group Health Cooperative,[11] has been proposed as a solution to improve the integrated management of chronic diseases. This model identifies six essential elements: community resources and policies, healthcare organisation, self-management support, delivery system design, decision support and clinical information systems. However, what constitutes an integrated chronic care model and how it is implemented and delivered within healthcare services, has continued to evolve with the advent of new interventions. Most of the chronic care models have been developed in high-income countries with sophisticated health services,[16] which may not appropriately apply to SSA.

The untapped capacity of chronic care models for HIV/AIDS care in SSA should be viewed as emerging models for other chronic diseases including cardiometabolic diseases.[17] Lessons learnt from the implementation of care models for HIV/AIDS and tuberculosis in SSA could be effectively applied to improve care models for cardiometabolic diseases.[18] A systematic review by Rohwer et al[19] examined existing integrated care models for diabetes and hypertension in LMICs. However, the included studies did not focus on structured clinical care delivered to persons living with multimorbidity. Furthermore, the integrated care for hypertension and diabetes was based on a 'one-stop-shop' model where all healthcare services are provided under one roof.[19] This is just one way of describing integrated care. A comprehensive evaluation of the applicability of the elements of chronic care models for c.ardiometabolic multimorbidity has not been systematically evaluated in SSA. Therefore, the purpose of this systematic review was to identify elements of integrated chronic care models for cardiometabolic multimorbidity in SSA and their effects on clinical or mental health outcomes including systolic blood pressure (SBP), blood sugar, depression scores and other patient-reported outcomes such as quality of life and medication adherence.

## METHODS

### Protocol registration

The study protocol was registered in the International Prospective Register for Systematic Reviews. The findings are reported according to the Preferred Reporting Items for Systematic Reviews and Meta-Analyses (PRISMA).[20]

### Eligibility criteria

#### Participants

We selected studies conducted in SSA among adult patients aged 18 years and over with cardiometabolic multimorbidity receiving care in a primary or community care setting. Cardiometabolic multimorbidity was defined as having two or more chronic conditions; at least one of which is a cardiometabolic disease such as type 2 diabetes, hypertension, hypercholesterolaemia, hypertriglyceridaemia, dyslipidaemia or CVDs such as stroke. Others included common discordant conditions such as HIV/AIDS, chronic respiratory diseases, cancers and mental illnesses. These conditions constitute over three-quarters of the global burden of chronic diseases.[21]

#### Intervention

We included interventions with integrated chronic care models for persons living with cardiometabolic multimorbidity. In line with previous studies,[22–26] we classified the intervention components into six categories based on the elements of Wagner's chronic care model.[11] Thus, for the inclusion criterion, the models were considered 'integrated care' if they applied at least two of the six chronic care model elements to manage more than one chronic disease.[23] This included self-management support, delivery system design, decision support, clinical information system, healthcare organisation and community linkages.

The classification of intervention features and components is shown in online supplemental file 1. Self-management support includes interventions that empower people living with chronic conditions to manage their health. Examples are self-help groups and technological aids for self-care. Delivery system design interventions focus on care delivery efficiency and effectiveness such as patient care planning, coordination and follow-up. Decision-support interventions promote clinical care that is consistent with scientific evidence and patient preferences. Clinical information systems is the organisation of patient data to facilitate efficient and effective care. Healthcare organisation interventions focus on creating an organisational culture and mechanisms that promote safe, high-quality care. Community linkages include interventions that mobilise community resources to meet the needs of people living with chronic diseases. Where the chronic care model had multiple components, we defined each element using the Wagner taxonomy[11] and highlighted the predominant components.

## Control

The control or comparison group in the included studies received the usual or standard care comprising stand-alone services for cardiometabolic multimorbidity.

## Outcome

We included studies that reported any objective measure of patient clinical or mental health outcomes. For example, SBP (mm Hg), glycated haemoglobin (HbA1c (%)) and depression scores. Others included patient-reported outcomes such as quality of life, and patient behaviour (medication adherence).

## Types of studies

We included randomised controlled trials (RCTs) meeting the quality criteria developed by the Cochrane Effective Practice and Organisation of Care (EPOC).[27] This is because RCTs provide more unbiased information than other study designs on the differential effects of chronic care models. Studies from SSA published in all languages between 1999 and December 2022 were included. We used 1999 because this was the formal year of inception of Wagner's chronic care model.[28]

## Exclusion criteria

We excluded single-disease-focused studies and studies conducted among pregnant women. Interventions with no specified structured clinical care delivered to persons living with cardiometabolic multimorbidity were excluded. Studies with interventions for comorbid conditions where the intervention was targeted solely at one of the conditions rather than integrated care for the co-occurring conditions were also excluded. Studies that assumed multimorbidity to be the norm based on individuals' age without screening and diagnosis by qualified healthcare providers were excluded as the interventions were not being targeted specifically at multimorbidity.

## Search strategy

Two reviewers independently searched PubMed, Embase, Scopus, Web of Science, Cumulative Index to Nursing and Allied Health Literature (CINAHL), Cochrane library, African Journals Online, Informit and PsycINFO without language restriction. The search was limited to articles indexed from 1999 up to 15 December 2022 to capture the beginning of the application of the elements of Wagner's chronic care model. The search results were limited by filters for RCTs and search concepts for chronic care models, cardiometabolic diseases and multimorbidity (see online supplemental file 2). Additional articles were identified by scanning the reference list of relevant studies obtained through the search. Handsearching of key journals was also conducted. Furthermore, we searched for registered trials in the ClinicalTrials.gov, the Pan African Clinical Trials Registry and grey literature from OpenSIGLE.

## Data collection and analysis

### Selection of studies

The search results were uploaded into the EndNote V.12 reference manager (Clarivate, Philadelphia, USA) and screened for duplicates. Four review authors (EW, MN, PK and DM) independently performed the initial screening of titles and abstracts. Results of screening were recorded against the citation in Excel spreadsheet. The articles selected for full-text review were subsequently assessed for adequacy following the inclusion criteria for patient, intervention, comparator and outcome. Consensus meetings were held between the reviewers and the project coordinator (PO) to resolve disagreements. We used a modified PRISMA flow chart to describe the study selection (figure 1).

### Data extraction and management

Four review authors (EW, MN, PK and DM) abstracted data using a modified version of the EPOC data collection checklist.[27] Data were entered in Excel spreadsheets and exported to Stata V.15 (StataCorp). Data quality disagreements were resolved by consensus between the review authors and the project coordinator (PO). Information extracted from the included studies comprised:

1. Participants: type of patients, age, nature of multimorbidity and how it was diagnosed.
2. Study design: randomisation, the unit of allocation and follow-up.
3. Intervention: we extracted a full description of the intervention components and features. This included self-management support, delivery system design, decision support, clinical information system, healthcare organisation and community resources.
4. Service providers: specialists, primary care providers and family members.
5. Study setting: primary care and community-based care.
6. Patient clinical or mental health outcomes such as SBP, blood glucose control and depression symptom scores. Others included patient-reported outcomes such as quality of life and medication adherence.
7. Results: we organised results into clinical outcomes, mental health outcomes and patient-reported outcomes.

### Assessment of risk of bias

Four review authors (EW, MN, PK and DM) independently assessed the risk of bias in the included studies using standard EPOC criteria.[29] The assessment domains included: allocation (sequence generation and concealment); baseline characteristics; incomplete outcome data; contamination; blinding; selective outcome reporting; and other potential sources of bias. The criteria for the assessment were 'low risk', 'high risk' or 'unclear risk' (figure 2). Loss to follow-up of more than 20% was considered a high risk of bias. We also considered loss to follow-up as a high risk of bias when it was unbalanced between the groups. Published protocols and trial registrations were tracked to assess selective reporting bias. We assessed incomplete

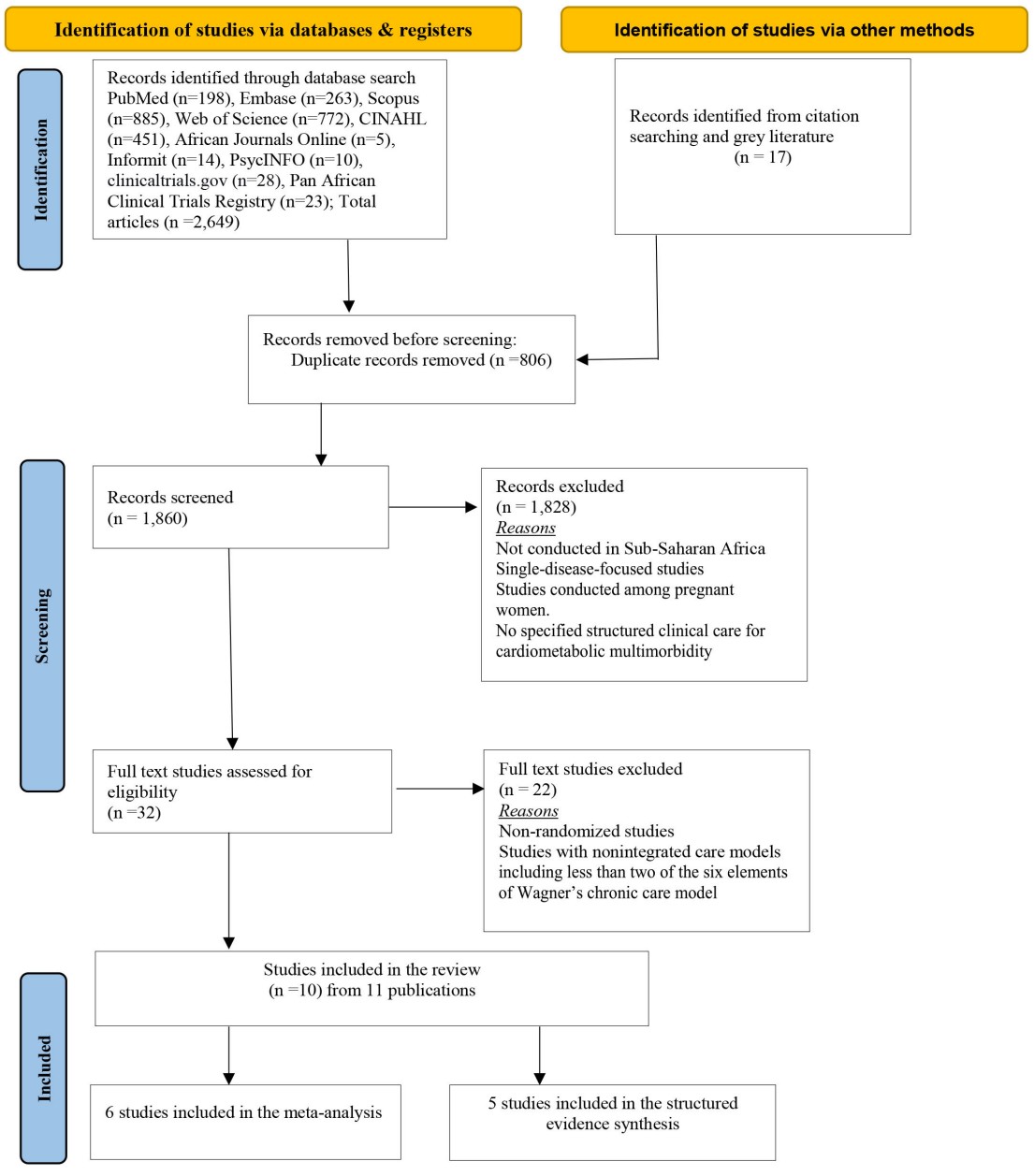

**Figure 1** Preferred Reporting Items for Systematic Reviews and Meta-Analyses flow diagram of the article selection process.

reporting for studies that reported different results than the outcomes outlined in the methods sections of selected articles. Any disagreements between the review authors were resolved by consensus, or with the consultation of the project coordinator (PO).

## Data analysis

We used a random-effects meta-analysis to estimate the pooled mean effect size of integrated care versus standard care on SBP. The random-effects meta-analysis allows for statistical heterogeneity between studies by assuming that the true effects in the individual studies differ from each other. Six studies with similarities in terms of the patient population, interventions and outcome assessment, were included in the meta-analysis. Studies with significant heterogeneity were not included in the meta-analysis. Hence, we conducted a structured synthesis of the results.

All analyses were performed using 'metan' command in Stata V.15 (StataCorp).

We calculated the effect size following the methods described by Hedges and Olkin.[30] For each of the included studies, we calculated results in terms of absolute and standardised mean differences with a 95% CI. The heterogeneity of the selected studies were evaluated using $\chi^2$ and $I^2$ statistics.[31] Using the $\chi^2$ test, heterogeneity between studies was considered significant if $p<0.10$.[32] We used $I^2$ statistics, values of 25%–49%, 50%–74% and >75% to determine low, moderate and high heterogeneity, respectively.[33] Publication bias was assessed using the Doi plot and Luis Furuya Kanamori (LFK) Index.[34] The LFK Index outside the interval −1 and 1 was considered consistent with asymmetry.[35] As Higgins *et al*[36] posit, it is justifiable to perform a pooled meta-analysis

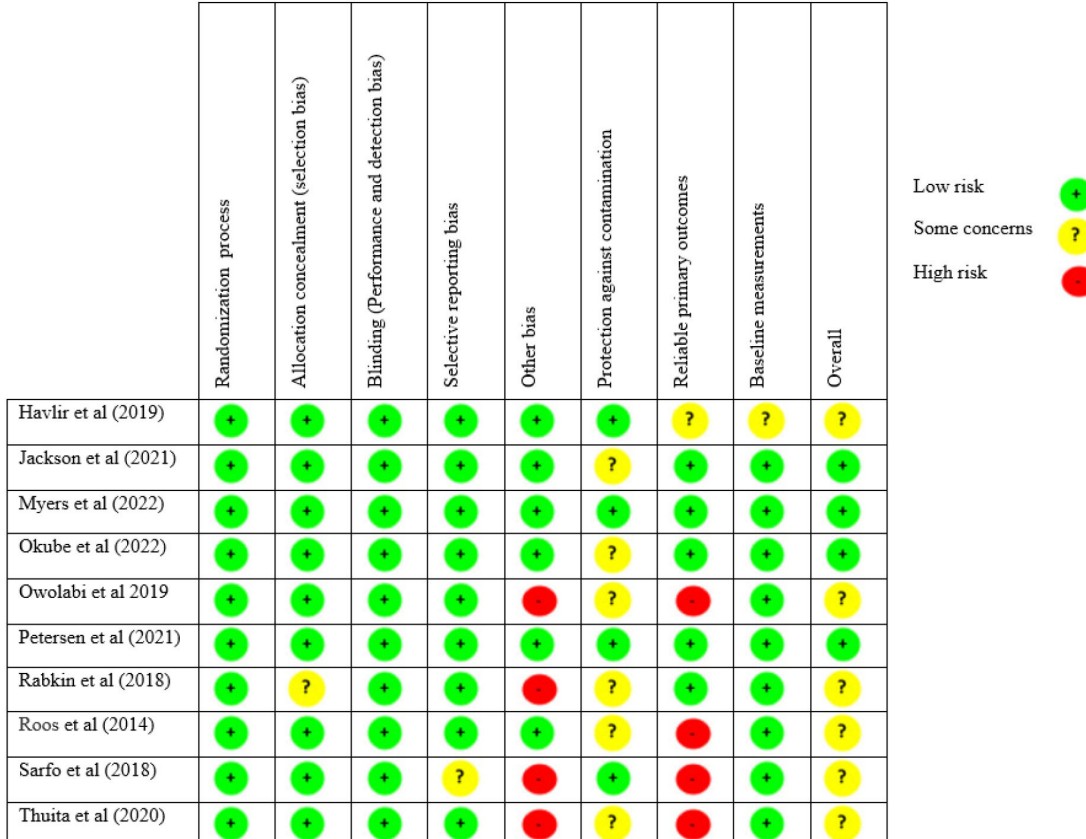

**Figure 2** Risk of bias in included studies.

involving cluster RCTs and individual RCTs. However, sensitivity analyses should be performed to test the robustness of the pooled estimates.[36 37] We performed sensitivity analyses to assess the robustness of our results by comparing the pooled effect estimates from separate analyses involving RCTs (four studies) and cluster RCTs (two studies).

### Certainty of evidence
The certainty of the evidence of included studies was assessed using the Grading of Recommendations Assessment, Development and Evaluation (GRADE) methodological guideline. According to GRADE guidance, RCTs start with high-certainty evidence.[38] Our judgements to downgrade the certainty of evidence were based on the assessment of the following five domains: study limitations, inconsistency, imprecision, indirectness and publication bias. For each outcome, we described the certainty of evidence as high, moderate, low or very low. The GRADE criteria also take into consideration the certainty of evidence and the size of the effect.[39] We considered a 5 mm Hg reduction in SBP to be clinically significant.[40] A change of 0.5% for the outcome HbA1c, and 10% for depression scores, medication adherence and quality of life were considered clinically significant.[41] For each outcome, we described the certainty of evidence as high, moderate, low or very low.[42]

### Patient and public involvement
Patients or the public were not involved in the design, conduct, reporting or dissemination of this research.

## RESULTS
### Search results
The PRISMA flow diagram of the article selection process is presented in figure 1. We identified 2649 studies from our searches. Another 17 studies were included after manually searching the reference list and grey literature. After the removal of 806 duplicate records, we screened the titles and abstracts of 1860 articles and identified 32 potentially eligible studies. We reviewed the full text of 32 studies and excluded 22 studies. Thus, 10 studies from 11 publications were included in this review (6 RCTs and 4 cluster randomised trials). Of these 10 studies, 6 were included in the meta-analyses and 4 studies from 5 publications were included in the structured synthesis of the results.

### Characteristics of included studies
The characteristics of the included studies are presented in table 1. We included 10 trials from 11 publications[43–53] with a total of 4864 participants. Sample sizes ranged from 55 to 1174 participants. Three studies were conducted in South Africa, two in Kenya and Nigeria and one in Eswatini and Ghana, respectively. There was one multicountry

**Table 1** Characteristics of included studies

| Study | Setting | Study design | Sample size | Age, years | Multimorbidity | Follow-up (months) | Outcome | Intervention |
|---|---|---|---|---|---|---|---|---|
| Havlir et al Uganda and Kenya[43] | Primary and community-based care | Cluster RCT | 1441 patients with hypertension and HIV | ≥30 | Hypertension and HIV | 36 | BP control | Multidisease testing; integrated care for HIV and hypertension; structured follow-up, flexible hours of operation and reduced wait time at clinics; and patient incentives. |
| Jackson and Ukwe Nigeria[44] | Primary care | RCT | 182 | 18–69 | Hypertension and HIV | 12 | SBP and medication adherence | Pharmaceutical care with structured education and counselling by a pharmacist after seeing a physician on their clinical visit. Education focused on self-monitoring of BP, lifestyle modification, reviewing the date of the next appointment and prescription. |
| Jackson and Ukwe Nigeria[45] | Primary care | RCT | 182 | 18–69 | Hypertension and HIV | 12 | Health-related quality of life | Structured education and counselling on general self-care, medicines use storage and lifestyle modifications. |
| Myers et al South Africa[46] | Primary care | Cluster RCT | 1174 | ≥18 | HIV, type 2 diabetes, depression and substance use disorder | 12 | Depression scores | Psychological interventions comprising motivational interviewing and problem-solving therapy delivered by a trained facility-based community health worker. |
| Okube et al Kenya[47] | Community-based care | RCT | 294 | 18–64 | Metabolic syndrome | 12 | BP and blood sugar control | Community-based health education on lifestyle modification and face-to-face delivery of verbal and written individualised health recommendations on risk factors for CVDs. |
| Owolabi et al Nigeria[48] | Primary care | RCT | 158 | ≥18 | Hypertension and stroke | 12 | SBP | A culturally appropriate, multipronged intervention comprising patient global risk factor control report card, personalised phone text-messaging and educational video. |
| Petersen et al South Africa[49] | Primary care | Cluster RCT | 925 | ≥18 | Hypertension and depression | 12 | Depression scores | Supplementary training of primary nurses and doctors on mental health and clinical communication skills. Collaborative care model for patients with hypertension and comorbid depressive symptoms including doctors, nurses, clinical psychologists and lay counsellors. |
| Rabkin et al Eswatini[50] | Primary care | RCT | 236 | ≥40 | HIV, hypertension, type 2 diabetes, and hyperlipidaemia | 6 | SBP and HbA1c | CVD risk factors screening and structured referrals among patients living with HIV. |
| Roos et al South Africa[51] | Primary care | RCT | 84 | 20–65 | HIV and metabolic diseases | 12 | SBP and fasting blood sugar | Pedometer and a physical activity diary with education materials and self-monitoring documents. Structured regular clinical sessions for review of physical activity diary and risk factors for ischaemic heart disease. Monthly SMS text motivational messages. |
| Sarfo et al (2018) Ghana[52] | Primary care | Cluster RCT | 55 | ≥18 | Hypertension and stroke | 9 | BP control and medication adherence | Bluetoothed BP device and smartphone for self-monitoring and reporting BP measurements and medication intake. Tailored motivational text messages delivered based on the levels of adherence to the medication intake protocol. |
| Thuita et al Kenya[53] | Primary care | RCT | 133 | 20–79 | Metabolic syndrome | 6 | SBP and HbA1c | Integrated care with nutrition education and peer-to-peer support. |

The control group received standard/usual care.
BP, blood pressure; CVDs, cardiovascular diseases ; HbA1c, glycated haemoglobin ; RCTs, randomised controlled trials; SBP, systolic blood pressure; SMS, short messaging service.

study conducted in Uganda and Kenya. Studies mostly took place in primary care settings (n=9) or both community and primary care settings (n=2). Most trials recruited participants with HIV and cardiometabolic diseases (n=9), another two recruited patients with HIV, type 2 diabetes, hypertension, depression and substance use disorder. Eight studies lasted 12 months or more. In all the included studies, the control group received the usual standard care while the intervention group received structured integrated care with varying elements of chronic care models including self-management support, delivery system design, decision support, clinical information system, healthcare organisation and community resources.

### Risk of bias in included studies

The risk of bias summary is presented in figure 2. All the included studies were RCTs. Hence, all studies had low risk for randomisation. Four of the 11 publications had a cluster design that ensured no contamination of control participants.[43 46 49 52] The four cluster-RCTs accounted for clustering effects in their analysis so there was no unit of analysis errors.[43 46 49 52] Six studies were appraised as unclear risk due to insufficient information about protection against contamination.[44 47 48 50 51 53] One study was rated as unclear risk due to lack of sufficient information about allocation concealment. Although it was not feasible to blind the participants and study personnel due to the nature of the interventions, the absence of blinding did not affect the objective outcomes. Hence, all studies were graded as low risk for blinding. Under the reliability of primary outcomes, five studies were rated as high risk due to small sample sizes[48 51–53] and lack of baseline measurements.[43] For selective reporting, all the studies were assessed as low risk, except one that lacked clarity and was assessed as unclear risk.[50] Four studies were rated high risk for other biases, mainly short follow-up duration (less than 1 year).[48 50 52 53] Overall, the certainties of the evidence for SBP, HbA1c, depression, medication adherence and quality of life were downgraded to moderate. This was due to the high risk of bias, imprecision and clinically insignificant effect sizes in the included studies.[43–53]

### Components of the chronic care model

Table 2 presents the components of the chronic care model in the included studies. All studies had to include at least two components of the Wagner chronic care model to be defined as 'integrated care' for the management of at least two chronic diseases. The number of chronic care elements ranged from 2 to 5. One study included five of the six elements of the chronic care model.[43] Three studies included three elements[48 51 52] and six enclosed two elements.[45–47 49 50 53] All the studies included delivery system design except one. The components of self-delivery included multidisease screening and treatment, task shifting, structured follow-up and collaborative care. Seven studies included self-management support such as home-based self-monitoring, lifestyle counselling,

post-clinical patient follow-up and patient support groups.[43 44 47 48 51–53] Four studies included decision support components such as supplementary training of healthcare workers and support supervision.[43 46 49 50] Three studies included clinical information systems.[48 51 52] The elements enclosed comprised hospital registries, patient diaries, report cards and electronic platforms for blood pressure (BP) monitoring. Two studies included community resource elements to enhance health campaigns.[43 47] One study included the healthcare organisation element, mainly quality improvement strategies to enhance access to medication and treatment services.[43]

### Effectiveness of integrated models of care compared with standard care

#### Systolic blood pressure

Figure 3 presents the random effects meta-analysis of SBP. The pooled results of six studies[44 47–49 52 53] involving 1754 participants show that integrated versus standard care conferred a lower mean SBP (mean difference (MD)=−4.85 mm Hg, 95% CI (−7.37 to −2.34) with a small Hedges' g effect size (g=−0.25, (−0.39 to −0.11). However, the overall quality of evidence based on GRADE criteria was moderate. Moderate heterogeneity ($I^2$=39.57%) existed in the included studies (figure 3). The results of the sensitivity analyses separating RCTs from cluster RCTs are consistent with those of the pooled analyses (online supplemental file 3). Integrated care reduced SBP in RCTs by (MD=−5.71 mm Hg, 95% CI −9.32 to −2.09) and by (MD=−3.30 mm Hg, 95% CI −5.97 to −0.63) for the cluster RCTs. A Doi plot of the random effects meta-analysis of SBP for integrated versus standard care is shown in online supplemental file 4. The presence of publication bias is suggested by the asymmetrical visual inspection of the Doi plot, confirmed by LFK Index=−2.37.

We did not conduct meta-analyses for the four of the included studies in this review,[43 46 50 51] as they were too heterogeneous in terms of patient population, interventions and outcome assessment. Hence, we conducted a structured synthesis of their results. Of the studies included in the structured synthesis of results, the effect of integrated versus standard care on BP were reported in three studies.[43 50 51] However, the quality of evidence based on GRADE criteria was moderate. The SEARCH trial conducted in Kenya and Uganda found that hypertension control was 22% higher among patients on integrated care versus standard care (relative prevalence=1.22; (95% CI 1.08 to 1.37)).[43] In contrast, a study by Rabkin et al[50] conducted in Eswatini found no significant effect of integrated versus referred management of CVD risk factors on mean SBP. Another study by Roos et al[51] also found no significant effect of education and a home-based pedometer walking programme on SBP MD=2.50 mm Hg, 95% CI (−2.78 to 7.78).

#### Fasting blood glucose and HbA1c

Two studies included in the structured synthesis of the results assessed the effect of integrated versus standard

**Table 2** Elements of the chronic care model included in the studies

| Author (year) | Self-management support | Delivery system design | Decision support | Clinical information system | Healthcare organisation | Community resources |
|---|---|---|---|---|---|---|
| Havlir et al (2019)[43] | Enhanced lifestyle modification counselling in primary care centres. | Point-of-care multidisease screening for HIV, hypertension and diabetes with structured follow-up and care linkages. | Telephone and in-person oversight from a senior physician on the services provided by general physicians and nurses. | | Quality improvement strategies such as guaranteed access to medication, flexible hours of operations and reduced wait time at clinics. | Multidisease testing community health campaigns using community resource persons. |
| Jackson and Ukwe (2021)[44] | Education on self-monitoring of BP, lifestyle modification, self-care and appropriate use of medicines. | Structured pharmaceutical care including prescription review and follow-up. | | | | |
| Myers et al (2022)[46] | | Task shifting and empowerment of community health workers to provide basic psychological interventions. | Trained community health workers on basic psychological intervention including motivational interviewing and problem-solving therapy. | | | |
| Okube et al (2022)[47] | Individualised health recommendations on CVD risk factors. | | | | | Community-based health education on lifestyle modification. |
| Owolabi et al 2019[48] | Post-clinical follow-up phone texts and waiting room educational video. | Enhanced follow-up visits and pre-appointment phone texts. | | Hospital registry and patient report card as part of medical chart. | | |
| Petersen et al (2021)[49] | | Collaborative care model for patients with hypertension and comorbid depressive symptoms including doctors, nurses, clinical psychologist and lay counsellors. | Supplementary training of primary care nurses and doctors on mental health and clinical communication skills. | | | |
| Rabkin et al (2018)[50] | – | One stop shop for CVD risk factors screening and structured referrals among patients living with HIV. | Training of HIV clinical nurses and doctors to conduct CVD risk factors screening during routine clinical appointments of patients. | | | |
| Roos et al (2014)[51] | A pedometer and a physical activity diary with education materials and self-monitoring documents. Monthly SMS text motivational messages. | Structured regular clinical sessions for review of physical activity diary and risk factors for ischaemic heart disease. | | Patient diary for self-monitoring of risk factors for ischaemic heart disease. | | |

Continued

**Table 2** Continued

| Author (year) | Self-management support | Delivery system design | Decision support | Clinical information system | Healthcare organisation | Community resources |
|---|---|---|---|---|---|---|
| Sarfo et al (2018)[52] | Self-monitoring of BP using a bluetoothed device. Tailored motivational text messages delivered based on the levels of adherence to medication. | Structured follow-up for BP measurements and medication adherence. | | Digital platform for tracking BP measurements and medication adherence. | | |
| Thuita et al (2020[53] | Nutrition counselling and peer support group. | Monthly follow-up visits and structured regular clinical sessions to review patient progress. Facility-based patient support groups. | | | | |

BP, blood pressure; CVD, cardiovascular disease ; SMS, short messaging service.

care on fasting blood sugar (FBS) and HbA1c.[47 50] A study by Okube et al[47] found that integrated care conferred a significant (p<0.05) reduction in FBS by −0.5 mmol/L versus +0.08 mmol/L among patients on standard care. In a study by Rabkin et al,[50] the mean of HbA1c was significantly reduced by −0.7% (95% CI −1.3% to −0.1%) and −1.4% (−2.5% to −0.2%) in the integrated and standard care arms, respectively, with no statistical difference between arms 0.7% (95% CI −0.4% to 1.8%).

### Depression
The effect of integrated versus standard care on depression was reported in two studies. A study by Petersen et al[49] found no significant difference in the depression scores in patients on integrated versus standard care (adjusted risk difference=−0.04 (95% CI −0.19 to 0.11)).

By contrast, the results of the study by Myers et al[46] found significant differences in the depression scores in patients on integrated versus standard care (MD=4.8 (95% CI −7.2 to −2.4)).

### Medication adherence and quality of life
Two studies assessed the effect of integrated care on hypertension medication adherence.[45 50] One study reported the effect of integrated versus standard care health-related quality of life (HRQoL).[45] In a study by Rabkin et al,[50] integrated care conferred a higher likelihood of hypertension medication adherence than standard care (relative risk=1.28 (95% CI 1.10 to 1.47)). A study by Jackson and Ukwe[45] found that integrated care led to significant improvements in hypertension medication adherence (mean difference in difference (DiD)=2.32;

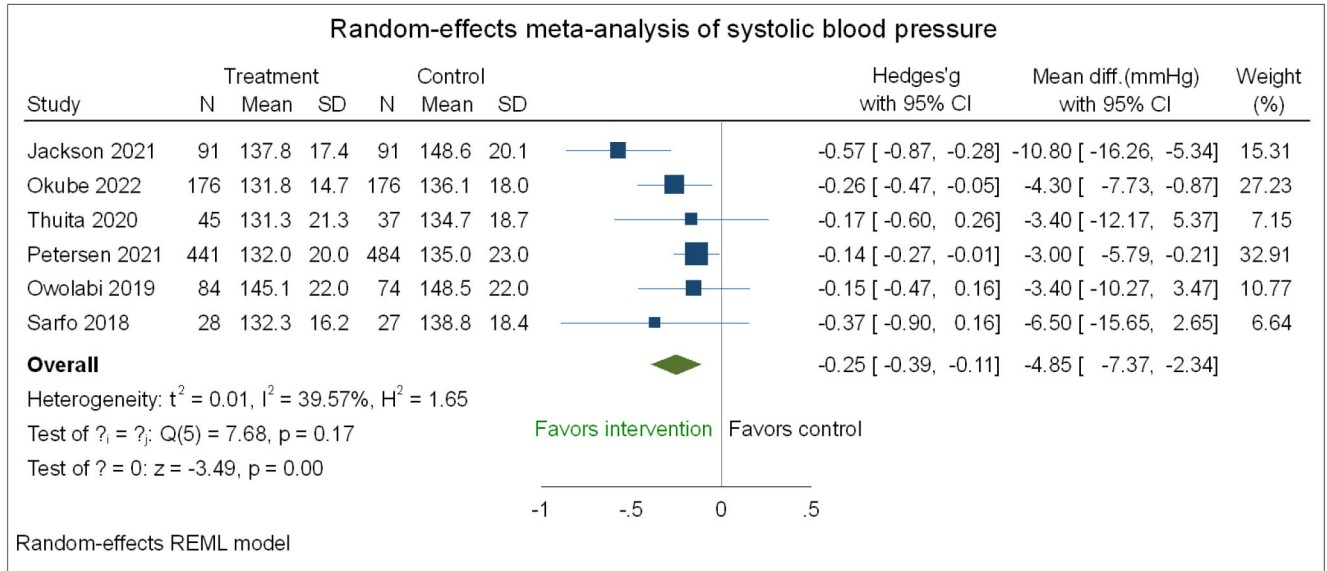

**Figure 3** Forest plot of the random effects meta-analysis of the effect of integrated versus standard care on systolic blood pressure for people with cardiometabolic multimorbidity in sub-Saharan Africa.

p<0.001) and HRQoL (DiD=6.5%, p<0.001). Two studies assessed the effect of integrated care on hypertension medication adherence.[45 50] One study reported the effect of integrated versus standard care on HRQoL.[45] In a study by Rabkin et al,[50] integrated care conferred a higher likelihood of hypertension medication adherence than standard care (relative risk=1.28 (95% CI 1.10 to 1.47)). A study by Jackson and Ukwe [45] found that integrated care led to significant improvements in hypertension medication adherence (mean DiD=2.32; p<0.001) and HRQoL (DiD=6.5%, p<0.001).

## DISCUSSIONS

This paper presents the results of a systematic review of integrated care models for cardiometabolic multimorbidity and their effects on intermediate health outcomes in SSA. Our results show that, in comparison to usual care, integrated care featuring at least two elements of Wagner's chronic care model conferred moderate improvements in SBP among patients living with cardiometabolic multimorbidity. The overall quality of evidence was moderate. Results of the systematic review suggest that integrated care compared with usual care has mixed results with regards to HbA1c, depression, medication adherence and quality of life, with some studies showing a significant effect and some no effect.

The findings of the current review based on RCTs are broadly consistent with reviews on the effectiveness of the integrated chronic care model from previous observational studies.[54 55] In contrast, one systematic review by Rohwer et al[19] on integrated care models for diabetes and hypertension in LMICs found no evidence of improved BP or diabetes control. However, the evidence presented had very low certainty and the included studies did not focus on care models for persons living with multimorbidity.

Our study used Wagner's chronic care model, a framework for improving the integrated management of chronic diseases through the implementation of its six core elements: community resources and policies, healthcare organisation, self-management support, delivery system design, decision support and clinical information systems.[11] A review by Goh et al[54] found greater improvements in HbA1c for care models with more elements over a single element. In the same vein, another systematic review by Ellisen et al [25] reported variations in glycaemic control with an increase in the number of elements of the chronic care model. However, most studies included in the current review had two to three CCM elements which could partly explain the mixed results with regard to the effect of care integration on intermediate health outcomes such as HbA1c, depression, medication adherence and quality of life, with some studies showing a significant effect and some no effect.

In the current review, delivery system design and self-management support were the most predominant components of the integrated chronic care models while healthcare organisation and community resource elements were less common. Nevertheless, the structured synthesis of the results shows no significant differences in health outcomes by the elements of care integration since a majority of studies included two to three elements. Previous studies have shown that intervention intensity including length of implementation and frequency of monitoring, rather than the number of intervention components may have an impact on the direction and magnitude of the health outcomes.[55 56]

Overall, our results have three main implications. First, our findings have great public health significance given that a 5 mm Hg reduction of SBP could reduce the risk of major cardiovascular events by about 13%.[57] Thus, incorporating integrated care into the implementation of Wagner's chronic care model may partly address the needs of people living with cardiometabolic multimorbidity. Second, this review provides crucial evidence on the applicability of chronic care models in SSA and unearths the components of chronic care models for people living with cardiometabolic multimorbidity. Last, most integrated care models sought to support self-management and delivery system design. However, a few included healthcare organisations and community resource elements. More interventions should seek to incorporate these elements to support integrated care.

### Strengths and limitations

The current review has three main strengths. First, all the included studies were RCTs. Hence, the findings are more unbiased than previous reviews on integrated care in LMICs that are mostly based on observational study designs.[19 58–60] Second, this study comprehensively extracted important constructs of Wagner's chronic care model, thus offering deeper insights into the effectiveness of its elements and delivering crucial evidence to researchers and health stakeholders for future improvement of integrated care models. The consideration of both communicable and non-communicable diseases in the review is also a strength due to the converging burdens of infectious and non-communicable diseases in SSA.[7]

The findings of this review should be interpreted cautiously due to a few limitations. First, the majority of the included studies did not classify the components of chronic care models in the interventions. Although the three reviewers independently classified the type of chronic care model elements in the studies using a standard guide, there is potential for misclassification bias. Second, the varying numbers and types of chronic care model components added to the heterogeneity. Furthermore, the absolute effect attributable to a particular element of the chronic care model remains unknown, as most included studies used multicomponent interventions. Third, it may not be possible to have a standard of care that is similar for all the studies included in this review. This resulted in a broad variety of usual care. Last, due to the substantial heterogeneity in the studies, in terms of multimorbidity, interventions, follow-up periods

and outcome assessment, only 6 of the 10 studies were included in the meta-analysis, which may have a bearing on the power of the meta-analysis findings. The rest of the studies were included in the structured synthesis of the results.

## Conclusions

We found that integrated care may lead to moderate improvements in SBP among patients living with cardiometabolic multimorbidity in SSA. This review highlights the paucity of research on care integration interventions for cardiometabolic multimorbidity in SSA since the selected RCTs were from only a few SSA countries. In addition, there were fewer studies with outcomes on HRQoL, mental health and medication adherence. The small number of studies addressing healthcare organisation and community resource elements provide less certainty on the benefits associated with these components. More studies are needed on the implementation effectiveness of integrated care models and its impact on cardiometabolic multimorbidity. The relative effectiveness of the different elements of integrated chronic care models and the cost-effectiveness of these models for the management of cardiometabolic multimorbidity should be explored. Other elements that are not well explored such as healthcare organisation and community resource elements should be investigated in future research. Future studies investigating the effectiveness of integrated care should classify the elements in the interventions and standardise the description of each element.

**Author affiliations**
[1]Chronic Disease Management Unit, African Population and Health Research Center, Nairobi, Kenya
[2]Department of Public & Occupational Health, Amsterdam UMC Locatie AMC, Amsterdam, The Netherlands
[3]Amsterdam Institute for Global Health and Development (AIGHD), AHTC, Amsterdam, Netherlands
[4]Research and Related Capacity Strengthening, African Population and Health Research Center, Nairobi, Kenya
[5]School of Health and Related Research, The University of Sheffield, Sheffield, UK
[6]African Network of Research Scientists, Nairobi, Kenya
[7]Emerging and Re-emerging infectious Diseases Unit, African Population and Health Research Center, Nairobi, Kenya
[8]Women's and Children's Health, Karolinska Institutet, Stockholm, Sweden

**Contributors** PO conceptualised the study, reviewed the literature and analysed the data. All authors contributed to the development and review of the protocol. EW, MN and DM screened titles and abstracts and participated in full-text screening; PO helped to resolve discrepancies. EW, MN and DM extracted data and assessed the risk of bias. PO assessed the certainty of evidence with input from all the authors. PO, JO and EW performed the meta-analyses. CA, HW and GA provided overall methodological guidance. JO and PK critically read and revised the first draft of the manuscript. PO, EW, PK and JO responded to reviewers' comments and made subsequent revisions. All authors have approved the final version of the manuscript. PO takes full responsibility for the work and/or the conduct of the study, had access to the data, and controlled the decision to publish.

**Funding** This systematic review was supported by funding from the Joep Lange Institute and in part by the Dutch Ministry of Foreign Affairs under the Joep Lange Chairs and Fellows Program. JO and AW are funded by Wellcome Trust (218462/Z/19/Z) doctoral training grant to the University of Sheffield. The funders

had no role in the design of the study and collection, analysis and interpretation of data and in writing the manuscript.

**Competing interests** None declared.

**Patient and public involvement** Patients and/or the public were not involved in the design, or conduct, or reporting, or dissemination plans of this research.

**Patient consent for publication** Not applicable.

**Provenance and peer review** Not commissioned; externally peer reviewed.

**Data availability statement** All data relevant to the study are included in the article or uploaded as supplementary information.

**ORCID iDs**
Peter Otieno http://orcid.org/0000-0001-6828-8301
Hesborn Wao http://orcid.org/0000-0002-6823-0895
Elvis Wambiya http://orcid.org/0000-0002-4149-3417

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
