## [Reviewer comments · BMJ Open]

ARTICLE DETAILS

TITLE (PROVISIONAL)	Effectiveness of integrated chronic care models for cardiometabolic multimorbidity in Sub-Saharan Africa: A systematic review and meta-analysis
AUTHORS	Otieno, Peter; Agyemang, Charles; Wao, Hesborn; Wambiya, Elvis; Ng'oda, Maurine; Mwangi, Daniel; Oguta, James; Kibe, Peter M.; Asiki, G

VERSION 1 – REVIEW

REVIEWER	Yuvaraj Krishnamoorthy JIPMER PSM
REVIEW RETURNED	21-Mar-2023

GENERAL COMMENTS	The authors have sincerely attempted to determine the effectiveness of integrated chronic care models for cardiometabolic multimorbidity in Sub-Saharan Africa. The review is methodologically robust and clearly reported per the latest PRISMA 2020 guidelines. The following are some of my suggestions to improve the manuscript further: 1) Though there are no strict guidelines, publication bias assessment using Funnel plot and Egger's test is usually recommended if there are at least 10 studies in the meta-analysis. This ensures enough power to detect publication bias or small study effects. Since only 6 studies were available for blood pressure outcome, it is better to cautiously interpret with the limitation or remove the publication bias assessment stating the limited number of studies.2) The following statement can be looked at carefully as both the studies show significant reduction, but the authors state there is no significant difference between the arms: "In a study by Rabkin et al [45] the mean of HbA1c was significantly reduced by -0.7% (95%CI: -1.3 to -0.1) and -1.4% (-2.5, -0.2) in the integrated and standard care arms, respectively with no statistical difference between arms"3) Mention systolic blood pressure instead of just blood pressure wherever the outcomes related to blood pressure are mentioned.4) Mention in the data analysis section that pooled mean difference will be reported for SBP outcome.5) An additional sensitivity analysis can be performed to check the robustness of the estimates6) Though components of GRADE are clearly mentioned and details related to study limitations and publication bias are provided, how decision-related to inconsistency, imprecision, and indirectness were made can be detailed (in the supplementary file if there are text restrictions).7) Discussion can provide additional insights about the study implications
---

REVIEWER	Joseph Kazibwe London School of Hygiene and Tropical Medicine Faculty of Public Health and Policy, Global Health and Development
REVIEW RETURNED	31-Mar-2023

GENERAL COMMENTS	The study focuses on elements of integrated care and their effect on some clinical health outcomes which is an area much needed in low- and middle-income countries considering the increasing prevalence of NCDs and thus multimorbidity. The study utilises recommended methodologies to address the aim of the study e.g systematic review, meta-analysis and structured synthesis. In addition, the quality assessment is robust, the authors assessed both uncertainty and the risk of bias among the included studies which were RCTs. The results are well aligned with the aims. This is paper presents results that will be essential in informing interventions targeting SSA in the future. Lines 111- 114: The authors mention the literature review that has been carried out in LMICs. It will be helpful for the reader if the authors mention some of the literature reviews that have been done on a global or international level too. Line 133: there is a typo. Delete z from “zas” Lines 130-132: the authors considered studies that includes patients receiving care at community or primary care level. However, no justification is provided. Explaining the reason is needed. Lines 180-187: Key words/terms for the search are missing, could you include the key words for the search? Lines 193-196: It is unclear how many researchers reviewed a given article, were they all the three? Lines 192-197: The authors used Endnote to identify duplicates but are unclear on what they used for screening articles. Did you use any platform for screening?
---

REVIEWER	Vainqueur Diakengua Centre de Recherche en Sante de Kimpese
REVIEW RETURNED	04-Apr-2023

GENERAL COMMENTS	Congratulations for the paper because it provide more researchs perspectives in how to implement integrated care for the chronic cardiometabolic disease in sub saharian Africa.
--

REVIEWER	Phuong Bich Tran University of Antwerp, Department of Family Medicine and Population Health
REVIEW RETURNED	27-Apr-2023

GENERAL COMMENTS	Overall, this study exhibits a high level of importance in investigating the effects of integrated care interventions for people with multimorbidity and exerts high quality in conducting systematic review and meta-analysis. Main comments  1. One concern regarding the meta-analysis based on only six
--

	studies is the potential limited generalizability of the findings. The small number of studies included may not adequately represent the diversity and complexity of integrated care interventions. Additionally, the limited number of studies may result in insufficient statistical power to detect small but potentially meaningful effects. Furthermore, the presence of heterogeneity among the included studies could impact the validity and reliability of the pooled effect size estimate. Therefore, cautious interpretation of the meta-analysis results is warranted, and further research with a larger number of studies is needed to strengthen the evidence base. 2. Was a homogeneity criteria checklist developed to enable the process of selecting studies for the meta-analysis? 3. How did the authors determine which aspect of the chronic care model is addressed by interventions that span across multiple aspects? For instance, how was it determined that the self-monitoring diary used in the study conducted by Roos et al. fulfilled both the self-management and clinical information system aspects of the model? 4. Certainty of Evidence: While the GRADE methodological guideline is mentioned for assessing the certainty of evidence, it would be beneficial to provide a brief description of how the assessment was conducted and any specific criteria used to determine the certainty level (high, moderate, low, or very low) for each outcome. This information would provide a better understanding of the strength of the evidence. 5. Line 393-397: “A review by Goh et al [51] found greater improvements in HbA1c for care models with more elements over a single element. However, most of the studies included in the current review had two to three elements. This could partly explain the mixed results with regards to effect of care integration on intermediate health outcomes such as HbA1c, depression, medication adherence and quality of life, with some studies showing a significant effect and some no effect.” This explanation lacks strength and clarity. While Goh et al. found that more than one element contributed to improved health outcomes, it fails to address why studies included in this review, which implemented more than 1 element (2-3 elements), still produced inconsistent results. The authors may consider rephrasing the statement in order to strengthen and clarify its meaning. Furthermore, it would be beneficial to delve further into the discussion regarding the optimal threshold for the number of elements to be included. 6. In addition, it would significantly enhance the paper if the discussion section incorporated implications related to future implementations and trials, as well as research and policy-making considerations concerning integrated care in similar settings. Minor comments 1. Line 133: Typo “zas” 2. In PRISMA chart: “5 studies included in the structured evidence synthesis”. In main text line 266: “4 studies were included in the structured synthesis of the results”.
--	--

	3. Line 368: Please change “integrated versus standard care HRQoL” to “integrated versus standard care on HRQoL”. 4. Line 375: Please change “This study presents the results of a systematic review” to “This paper presents the results of a systematic review” or “This review explored...”. 5. Line 383-384: “The findings of the current review based on RCTs are broadly consistent with reviews on the effectiveness of the integrated chronic care model from pervious observational studies.” Which parts of the findings are consistent and how? Please be more specific. 6. Line 384: Please change “pervious” to “previous”. 7. Line 387: Please remove “that”. 8. Line 395: Please change: “with regards to effect” to “with regards to the effect”. 9. Line 403: Please elaborate on the phrase “intervention intensity”.
--	---

VERSION 1 – AUTHOR RESPONSE

Reviewer: 1

Dr Yuvaraj Krishnamoorthy, JIPMER PSM

Comments to the Author.

The authors have sincerely attempted to determine the effectiveness of integrated chronic care models for cardiometabolic multimorbidity in Sub-Saharan Africa. The review is methodologically robust and reported per the latest PRISMA 2020 guidelines. The following are some of my suggestions to improve the manuscript further:

1. Though there are no strict guidelines, publication bias assessment using the Funnel plot and Egger's test is usually recommended if there are at least 10 studies in the meta-analysis. This ensures enough power to detect publication bias or small study effects. Since only 6 studies were available for blood pressure outcome, it is better to cautiously interpret with the limitation or remove the publication bias assessment stating the limited number of studies.

Thank you very much for this observation. Indeed concerns have been raised about both the visual appearance of funnel plots and the sensitivity of Egger's regression to detect asymmetry, particularly when the number of studies is small [4]. We have revised the publication bias assessment in the current review (Supplementary File 4). We used the Doi plot to visualize asymmetry and the LFK index to detect and quantify the asymmetry of the study effect as the number of studies included in the meta-analysis was small [5]. Lines 251 to 252

2. The following statement can be looked at carefully as both the studies show a significant reduction, but the authors state there is no significant difference between the arms: "In a study by Rabkin et al [45] the mean of HbA1c was significantly reduced by -0.7% (95%CI: -1.3 to -0.1) and -1.4% (-2.5, -0.2) in the integrated and standard care arms, respectively with no statistical difference between arms"

Thanks for pointing this out. Further estimates on the difference in the HbA1c reduction between the arms of the intervention and the treatment arm has been added (line 387).

3. Mention systolic blood pressure instead of just blood pressure wherever the outcomes related to blood pressure are mentioned.

Thank you very much for this observation “blood pressure” has been replaced with “systolic blood pressure” in the text

4. Mention in the data analysis section that pooled mean difference will be reported for SBP outcome.

Thank you very much for pointing out this. This has been added in lines 246-248.

5. An additional sensitivity analysis can be performed to check the robustness of the estimates

Thank you very much for this comment. Additional sensitivity analysis has been performed to check the robustness of the estimates. See lines 254-259 methods section and lines 357-360 in the results section.

6. Though components of GRADE are clearly mentioned and details related to study limitations and publication bias are provided, how decision-related to inconsistency, imprecision, and indirectness were made can be detailed (in the supplementary file if there are text restrictions).

Thank you very much for pointing out this. According to GRADE guidance, RCTs start with high-certainty evidence [6]. The quality of evidence was not very variable across outcomes. In lines 308-310 under the risk of bias of included studies, we have indicated that, the certainties of the evidence for SBP, HbA1c, depression, medication adherence and quality of life were downgraded to moderate. This was due to the high risk of bias, imprecision and clinically insignificant effect sizes in the included studies.

7. Discussion can provide additional insights about the study's implications.

Thank you very much for this observation. This has been added to lines 448 to 456

Reviewer: 2

Mr Joseph Kazibwe, London School of Hygiene and Tropical Medicine Faculty of Public Health and Policy

Comments to the Author:

The study focuses on elements of integrated care and their effect on some clinical health outcomes which is an area much needed in low- and middle-income countries considering the increasing prevalence of NCDs and thus multimorbidity. The study utilises recommended methodologies to address the aim of the study e.g. systematic review, meta-analysis and structured synthesis. In addition, the quality assessment is robust, the authors assessed both uncertainty and the risk of bias among the included studies which were RCTs. The results are well aligned with the aims. This paper presents results that will be essential in informing interventions targeting SSA in the future.

1. Lines 111- 114: The authors mention the literature review that has been carried out in LMICs. It will be helpful for the reader if the authors mention some of the literature reviews that have been done on a global or international level too

Thank you very much for pointing out this. The literature on chronic care models from global and regional perspectives has been highlighted in the background section (lines 113 to 125).

2. Line 133: there is a typo. Delete z from “zas”

This has been revised

3. Lines 130-132: the authors considered studies that includes patients receiving care at the community or primary care level. However, no justification is provided. Explaining the reason is needed.

Thank you very much for pointing out this. The Chronic Care Model is a framework developed to redesign care delivery for individuals living with chronic diseases in primary care [7]. A comprehensive evaluation of the applicability of the elements of chronic care models for cardiometabolic multimorbidity has not been systematically evaluated in SSA. Therefore, the purpose of this systematic review was to identify elements of integrated chronic care models for cardiometabolic multimorbidity in SSA and their effects on clinical or mental health outcomes including systolic blood pressure (SBP), blood sugar, depression scores and other patient-reported outcomes such as quality of life, and medication adherence.

4. Lines 180-187: Key words/terms for the search are missing, could you include the key words for the search?

Thank you for pointing out this. We have added the search syntax as a supplement document due to its length. See supplementary file 2.

5. Lines 193-196: It is unclear how many researchers reviewed a given article, were they all the three?

Thank you for highlighting this. In line 206, we have indicated that four review authors independently performed the initial screening of titles and abstracts.

7. Lines 192-197: The authors used Endnote to identify duplicates but are unclear on what they used for screening articles. Did you use any platform for screening?

Thank you very much for pointing out this. This has been revised. In line 200, we have indicated that the results of screening were recorded against the citation in an Excel spreadsheet.

Reviewer: 3

Vainqueur Diakengua, Centre de Recherche en Sante de Kimpese

Comments to the Author:

Congratulations for the paper because it provides more research perspectives on how to implement integrated care for chronic cardiometabolic disease in sub-Saharan Africa.

Thank you for this feedback. Indeed this review provides crucial evidence on the applicability of integrated chronic care models in SSA and unearths the components of chronic care models for people living with cardiometabolic multimorbidity

Reviewer: 4

Ms. Phuong Bich Tran, University of Antwerp

Comments to the Author:

Overall, this study exhibits a high level of importance in investigating the effects of integrated care interventions for people with multimorbidity and exerts high quality in conducting systematic review and meta-analysis.

Main comments

1. One concern regarding the meta-analysis based on only six studies is the potentially limited generalizability of the findings. The small number of studies included may not adequately represent the diversity and complexity of integrated care interventions. Additionally, the limited number of studies may result in insufficient statistical power to detect small but potentially meaningful effects. Furthermore, the presence of heterogeneity among the included studies could impact the validity and reliability of the pooled effect size estimate. Therefore, a cautious interpretation of the meta-analysis results is warranted, and further research with a larger number of studies is needed to strengthen the evidence base.

Thank you very much for these observations. We have added that the findings of this review should be interpreted cautiously due to the aforementioned limitations by the reviewer (lines 470-481 and 486-496). Nevertheless, the current review provides a comprehensive understanding of the applicability of the elements of chronic care models for cardiometabolic multimorbidity that has not been systematically evaluated in SSA

2. Was a homogeneity criteria checklist developed to enable the process of selecting studies for the meta-analysis?

Thank you very much for pointing this out. Four review authors (EW, MN, PK and DM) abstracted data using a modified version of the EPOC data collection checklist [8] (lines 206-209). Information extracted from the included studies is included in lines 211-112. Six studies with similarities in terms of the patient population, interventions, and outcome assessment, were included in the meta-analysis.

3. How did the authors determine which aspect of the chronic care model is addressed by interventions that span across multiple aspects? For instance, how was it determined that the self-monitoring diary used in the study conducted by Roos et al. fulfilled both the self-management and clinical information system aspects of the model?

Thank you very much for this observation. The classification of intervention features and components is shown in Supplementary File 1. Where the chronic care model had multiple components, we defined each element using the Wagner taxonomy [9] and highlighted the predominant components.

4. Certainty of Evidence: While the GRADE methodological guideline is mentioned for assessing the certainty of the evidence, it would be beneficial to provide a brief description of how the assessment was conducted and any specific criteria used to determine the certainty level (high, moderate, low, or very low) for each outcome. This information would provide a better understanding of the strength of the evidence.

Thank you very much for pointing out this. In response to the 1st reviewer's comment number 6, we have indicated that the quality of evidence was not very variable across outcomes. RCTs start with high-certainty evidence, according to GRADE guidance [6]. In lines 308-310 under the risk of bias of included studies, we have indicated that, the certainties of the evidence for SBP, HbA1c, depression, medication adherence and quality of life were downgraded to moderate. This was due to the high risk of bias, imprecision and clinically insignificant effect sizes in the included studies.

5. Line 393-397: "A review by Goh et al [51] found greater improvements in HbA1c for care models with more elements over a single element. However, most of the studies included in the current review had two to three elements. This could partly explain the mixed results with regards to the effect of care integration on intermediate health outcomes such as HbA1c, depression, medication adherence and quality of life, with some studies showing a significant effect and some no effect." This explanation lacks strength and clarity. While Goh et al. found that more than one element contributed to improved health outcomes, it fails to address why studies included in this review, which

implemented more than 1 element (2-3 elements), still produced inconsistent results. The authors may consider rephrasing the statement to strengthen and clarify its meaning. Furthermore, it would be beneficial to delve further into the discussion regarding the optimal threshold for the number of elements to be included.

Thank you very much for these observations. We have reviewed the phrase and added another reference to explain our results in lines 429 to 437.

6. In addition, it would significantly enhance the paper if the discussion section incorporated implications related to future implementations and trials, as well as research and policy-making considerations concerning integrated care in similar settings.

Thank you very much for this observation. The study implications have been added to lines 448 to 456

Minor comments

7. Line 133: Typo “zas”

This has been revised

8. In the PRISMA chart: “5 studies included in the structured evidence synthesis”. In main text line 266: “4 studies were included in the structured synthesis of the results”.

Thank you very much for this observation. The search results has been revised. In lines 279 to 285, we have indicated that 10 studies from 11 publications were included in this review (6 randomized controlled trials and 4 cluster randomized trials). Of these 10 studies, 6 were included in the meta-analyses and 4 studies from 5 publications were included in the structured synthesis of the results.

9. Line 368: Please change “integrated versus standard care HRQoL” to “integrated versus standard care on HRQoL”.

This has been revised

10. Line 375.: Please change “This study presents the results of a systematic review” to “This paper presents the results of a systematic review” or “This review explored...”.

This has been revised

11. Line 383-384: “The findings of the current review based on RCTs are broadly consistent with reviews on the effectiveness of the integrated chronic care model from pervious observational studies.” Which parts of the findings are consistent and how? Please be more specific.

Thank you for pointing this out. We have reviewed and added specific reference to HBA1c and SBP

12. Line 384: Please change “pervious” to “previous”.

This has been revised

13. Line 387: Please remove “that”.

This has been revised

14. Line 395: Please change: “with regards to effect” to “with regards to the effect”.

This has been revised

15. Line 403: Please elaborate on the phrase “intervention intensity”.

Thank you for this comment. In this context, intervention intensity was in reference to strength of intervention including length of implementation as was referred by Goh et al [10]. An explainer has been inserted in the text.

VERSION 2 – REVIEW

REVIEWER	Yuvaraj Krishnamoorthy JIPMER PSM
REVIEW RETURNED	29-May-2023
GENERAL COMMENTS	Thank you for addressing all the comments. No further comments from my side
REVIEWER	Joseph Kazibwe London School of Hygiene and Tropical Medicine Faculty of Public Health and Policy, Global Health and Development
REVIEW RETURNED	29-May-2023
GENERAL COMMENTS	No further comments
REVIEWER	Phuong Bich Tran University of Antwerp, Department of Family Medicine and Population Health
REVIEW RETURNED	05-Jun-2023
GENERAL COMMENTS	In this second round of review, I have no further comments. Congratulations to the team of authors.